# EPINEST, an agent-based model to simulate epidemic dynamics in large-scale poultry production and distribution networks

Francesco Pinotti[1]*, José Lourenço[2], Sunetra Gupta[1], Suman Das Gupta[3,4], Joerg Henning[3], Damer Blake[5], Fiona Tomley[5], Tony Barnett[5,6], Dirk Pfeiffer[5,7], Md. Ahasanul Hoque[8], Guillaume Fournié[5,9,10]

1 University of Oxford, Oxford, United Kingdom, 2 Católica Biomedical Research, Católica Medical School, Universidade Católica Portuguesa, Lisbon, Portugal, 3 School of Veterinary Science, The University of Queensland, Queensland, Australia, 4 Gulbali Institute, Charles Sturt University, Wagga Wagga, NSW, Australia, 5 Royal Veterinary College, London, United Kingdom, 6 The Firoz Lalji Centre for Africa, London School of Economics and Political Science, London, United Kingdom, 7 City University of Hong Kong, Hong Kong SAR, Hong Kong, 8 Chattogram Veterinary and Animal Sciences University, Chittagong, Bangladesh, 9 INRAE, VetAgro Sup, UMR EPIA, Université de Lyon, Marcy l'Etoile, 69280, France, 10 INRAE, VetAgro Sup, UMR EPIA, Université Clermont Auvergne, Saint Genès Champanelle, 63122, France

* francesco.pinotti@biology.ox.ac.uk

**Data Availability Statement:** Data and code required to generate and analyse simulation output are available at https://github.com/francescopinotti92/EPINEST.

## Abstract

The rapid intensification of poultry production raises important concerns about the associated risks of zoonotic infections. Here, we introduce EPINEST (EPIdemic NEtwork Simulation in poultry Transportation systems): an agent-based modelling framework designed to simulate pathogen transmission within realistic poultry production and distribution networks. We provide example applications to broiler production in Bangladesh, but the modular structure of the model allows for easy parameterization to suit specific countries and system configurations. Moreover, the framework enables the replication of a wide range of eco-epidemiological scenarios by incorporating diverse pathogen life-history traits, modes of transmission and interactions between multiple strains and/or pathogens. EPINEST was developed in the context of an interdisciplinary multi-centre study conducted in Bangladesh, India, Vietnam and Sri Lanka, and will facilitate the investigation of the spreading patterns of various health hazards such as avian influenza, *Campylobacter*, *Salmonella* and antimicrobial resistance in these countries. Furthermore, this modelling framework holds potential for broader application in veterinary epidemiology and One Health research, extending its relevance beyond poultry to encompass other livestock species and disease systems.

## Author summary

Poultry meat is important for improving nutrition in developing countries. However, the rapid growth of poultry production raises concerns about the risks of diseases that can be passed from animals to humans and cause outbreaks. To understand and manage these risks, we developed EPINEST, an agent-based modelling framework that allows investigating how diseases can spread within the networks of poultry farms, markets and their

**Funding:** F.P., S.G., J.H., D.B., F.T., T.B., D.P., M.A. H. and G.F. are supported by the UKRI GCRF One Health Poultry Hub (Grant No. B/S011269/1), one of twelve interdisciplinary research hubs funded under the UK government's Global Challenge Research Fund Interdisciplinary Research Hub initiative. G.F. is supported by the French National Research Agency and the French Ministry of Higher Education and Research. The funders had no role in study design, data collection and analysis, decision to publish, or preparation of the manuscript.

**Competing interests:** The authors have no competing interests to declare.

associated transportation systems. EPINEST can be adjusted to match the way in which poultry are raised and traded in specific countries. It considers different traits of pathogens, how they are transmitted, and how different strains or pathogen types can interact. While EPINEST was primarily developed to simulate the transmission of zoonotic pathogens (namely avian influenza, *Campylobacter*, *Salmonella* and other bacteria carrying resistance genes) in poultry populations in South and Southeast Asia, this modelling framework can also be useful for studying the transmission of other pathogens in other livestock species. EPINEST will help understand how poultry farming and trading shape pathogen spread, maintenance and evolution, and support decision-making to make poultry production safer and more sustainable.

## Introduction

Animal populations act as reservoirs for a wide range of zoonotic pathogens, such as Ebola virus, MERS-CoV, SARS-CoV-2, avian influenza viruses (AIVs), *Campylobacter* and *Salmonella* [1–6]. Within this context, livestock production is known to promote the risk of zoonotic infections [7]. In the case of emerging pathogens of wildlife, livestock may become intermediate or amplifier hosts, increasing odds of spillover into the human population [8]. The ongoing global intensification of livestock production raises critical questions about the role of husbandry and animal trading practices in shaping the risk of zoonotic epidemics or spillover events. [9, 10]. Unfortunately, however, a comprehensive understanding of how such risk is modulated and amplified along production and distribution networks (PDNs) is lacking.

Poultry production has become the fastest growing livestock sector in the last three decades, with rapid intensification occurring in low- and middle-income countries (LMICs) and particularly in South and Southeast Asia [11]. In many of these countries, intensive production did not replace local farming and trading practices completely, resulting in multiple modes of production and distribution articulated in ways that are poorly understood and which vary according to market and other conditions. While such transformative changes have proven instrumental towards improving food security, nutrition and economic and societal development e.g. in China, India, Bangladesh among others, they also require careful monitoring and investigation. Indeed, the growth of poultry production and distribution networks has brought novel challenges in terms of disease management: intensive farming, limited surveillance infrastructure and veterinary services and in many examples poor biosecurity conditions [12, 13] can lead to an environment replete with health hazards. For example, widespread sub-optimal use of antimicrobial drugs by poultry farmers represents a leading driver of the emergence of antimicrobial resistance [14–16].

In many LMICs, people prefer to obtain their poultry from live bird markets (LBMs), which are a longstanding feature of poultry trade and of urban and rural life. Within poultry PDNs, LBMs may be considered as hubs, sites wherein large numbers of people, and critically birds, meet and mix [17, 18]. Thus, they are major hotspots of AIV amplification and evolution [19], and have been implicated in sustaining viral transmission in domestic poultry [20]. The diverse ecology of AIV strains circulating within LBMs in Asia has been documented extensively [21–24]. Low pathogenic strains such as H9N2 AIV are commonly found among LBMs in Bangladesh, often at higher rates than in surrounding farms [25–27]. Since its first identification in 1996, highly-pathogenic H5N1 influenza has been detected in LBMs in many Asian countries [28–32].

While the biological risks within poultry production systems are widely acknowledged, they remain poorly characterised. This is partly due to the inherent complexity of PDNs, which makes it difficult to understand how such risks are modulated and increased along poultry value chains. Previous modelling efforts have focused on disease transmission within specific PDN settings, e.g. single farms or LBMs [33, 34], or some PDN segment, such as networks of farms or LBMs [18, 35–37]. Attempts to account for poultry or livestock PDN structure in infectious disease modelling are rare and mostly theoretical, often leaving out many epidemiologically relevant details of poultry production and distribution [38, 39]. Recent PDN mapping efforts have provided a clearer picture of PDNs in several Asian countries [40, 41]. A central observation is that PDNs are highly heterogeneous across countries, poultry types, and even within the same country. Therefore, a better understanding can be achieved by extending and developing modelling to increase our understanding of structural heterogeneities within and across PDNs.

To address this gap, we introduce EPINEST, a novel agent-based model (ABM) that allows simulation of pathogen transmission on top of realistic, empirically derived assumptions about poultry movements. EPINEST generates synthetic PDNs consisting of the key nodes, e.g. farms, traders, LBMs, that are responsible for the production and transportation of chickens through the PDN until they are sold to end-point consumers. Extensive data about farming and trading practices, collected mainly from field surveys, is used to inform PDN generation and simulation [18, 27]. Farm-specific data, for example, include farm locations, capacity and statistics of distinct stages of production cycles. Trader-level data encompass details of purchases and sales involving individual actors, origins of purchased poultry, and trader movements. EPINEST allows for substantial flexibility for users in terms of specifying PDN structure and functioning, making it a suitable framework to carry both data-driven and more open-ended analyses. In fact, the ABM permits customisation of many PDN properties, thus allowing users to explore a wide range of hypothetical PDN configurations.

This ABM provides a unified and flexible modelling framework to simulate epidemic dynamics in poultry PDNs and is the outcome of a wider interdisciplinary research initiative [42]. Within this context, EPINEST will enable investigating the amplification and dissemination of a wide range of health hazards, including AIV, *Campylobacter* and anti-microbial resistance genes in poultry systems in Bangladesh, India, Vietnam and Sri Lanka. More broadly, our framework may also be tailored to distinct poultry and livestock production realities to tackle a wider range of epidemiological questions.

In this paper, we provide a detailed description of our ABM and illustrate how to use it to explore a range of PDN structures and to better understand aspects of pathogen transmission in PDNs. The examples presented here are based on a broiler (chickens reared for meat) PDN in Bangladesh, which has been characterised extensively [18, 40], while epidemic simulations focus on the paradigmatic case of AIV transmission. The latter also illustrate an important feature of our framework, namely the ability to simulate multiple co-circulating pathogens and their interactions.

## Results

### Synthetic poultry networks

To address questions about the eco-epidemiological dynamics of AIVs and other poultry-related pathogens, we implemented an agent-based model to simulate pathogen transmission on top of synthetic PDNs. Within our framework, generated PDNs consists of four main types of nodes: farms, middlemen, vendors and LBMs (Fig 1A). The system works as a supply chain where chickens are reared in farms starting from day-old chicks and are later transported to

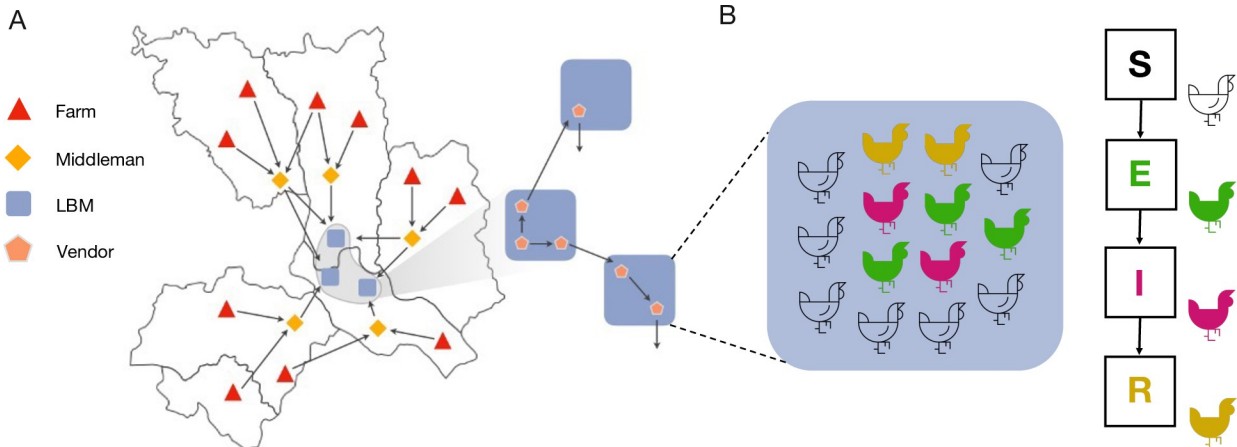

**Fig 1. Model schematics.** (A) Synthetic PDN and poultry movements. Chickens are produced in farms (red) across the study area, and transported to LBMs (blue) by middlemen (yellow). These are mobile traders that may collect chickens from multiple farms located in one or more upazilas/sub-districts (an administrative area below that of a district in Bangladesh). Within LBMs, chickens are handled by vendors (orange) and may be moved between LBMs as a result of vendors' trading practices. (B) Individual settings associated with farms, middlemen, LBMs (when open) and vendors (overnight, when LBMs are closed) provide the context for pathogen transmission, under the assumption that chickens mix homogeneously within the same setting. The panel zooms in on a single LBM, where chickens are colour-coded according to disease status: susceptible (S), exposed or latent (E), infectious (I) and recovered or immune (R). The base layer of the map was obtained from https://gadm.org/download_country_v2.html.

LBMs by middlemen (more details can be found in the Materials and methods section and in S1 Text). Once they arrive at the LBM stage, chickens are handled by vendors. These vendors may then sell chickens to other vendors operating in the same or different LBMs, and/or to endpoint consumers, in which case chickens are removed from the PDN. At any stage where chickens are exchanged, other than to the endpoint customer, an opportunity arises for pathogen exchange and mixing.

To illustrate the ability of the model to synthesize realistic poultry movements, we simulate a small PDN consisting of 1200 farms scattered across the 50 upazilas (sub-districts) that supply the largest amount of broiler chickens to LBMs located in Dhaka (Fig 2A). The simulated PDN includes 20 distinct LBMs, 163 middlemen and 444 vendors, and allows the trade of chickens between LBMs. Numbers of middlemen and vendors can not be specified a priori; instead, they are determined dynamically by initially calculating the average number of chickens that are sold by farms to each LBM daily. These calculations depend on the spatial arrangement of farms, their sizes and frequency of selling, i.e. parameters that can be specified a priori. The capacity of each trader (middleman or vendor), i.e. the maximum amount of chickens that he/she can purchase daily is also fixed over the course of a simulation.

Farms sell all their chickens at the end of a production cycle. The trading phase may require multiple days to complete and the flock may be split into multiple transactions involving different middlemen. Fig 2B and 2C show that both the distributions of farm trading times and numbers of transactions per production cycle obtained through simulations are consistent with field observations. Upstream transportation and distribution of poultry operated by middlemen represent an important driver of poultry mixing in LBMs [18]. In simulations, middlemen direct previously purchased chickens to LBMs depending on where these have been sourced from. In practice, a chicken bought in upazila $a$ is sold in market $l$ with probability $f_{a,l}$, as estimated from field questionnaires. Fig 2D shows that the ABM generates poultry fluxes between individual upazilas and LBMs that are in excellent agreement with the corresponding

 EPINEST, an agent-based model for epidemic simulations in poultry populations

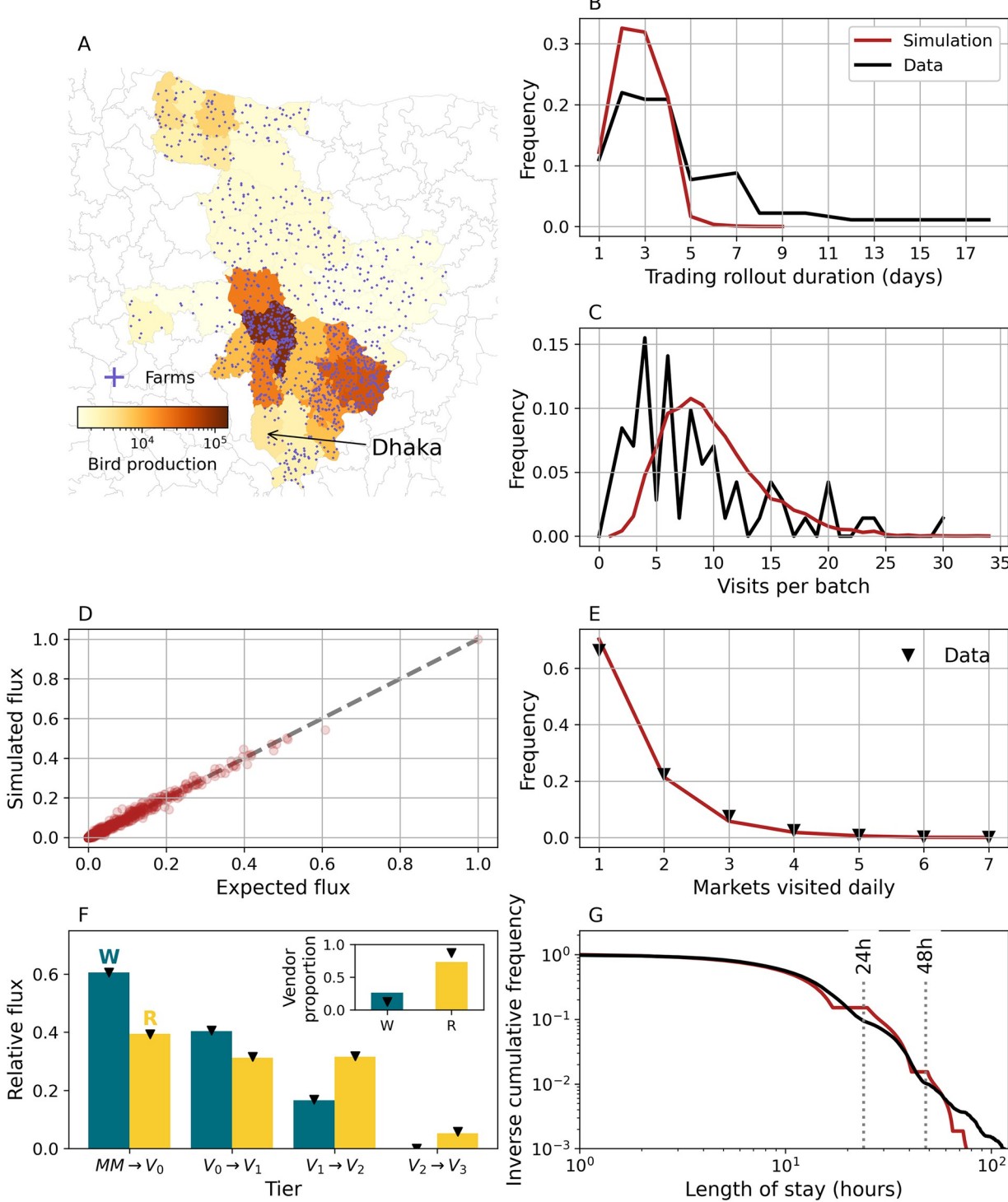

**Fig 2. Simulating poultry movements.** (A) Spatial population of 1200 farms supplying Dhaka. Farm locations are generated as described in S1 Text and assigned preferentially to upazilas with a larger observed outgoing chicken flux (colour scale). (B) Empirical (black) and simulated (red) distribution of times required to sell an entire batch. (C) Expected and measured distributions of transactions a single batch is split into. (D) Measured vs expected relative flux between individual pairs (dots) of upazilas and LBMs. (E) Distribution of LBMs serviced daily by individual middlemen. (F) Proportion of chickens sold to wholesalers (W, teal) and retailers (R, yellow) by LBM tier in simulations (bars) and data (markers). $MM \rightarrow V_0$ refers to transactions involving middlemen and first tier vendors, while $V_L \rightarrow V_{L+1}$ represents inter-tier transactions. For each tier, bars do not add up to 1 since wholesalers can sell to end-point consumers as well. Inset shows proportions of wholesalers and retailers. (G) Marketing time distribution. Results are obtained from a single simulation with default settings. We emphasize that some of the quantities

shown here (panels B,C,G), emerge dynamically during simulations and are not enforced as tightly as poultry fluxes (D) and visits to LBMs (E). Farm data are obtained from [27]. Data about middlemen and vendor trading practices and marketing times are obtained from [18]. The base layer of the map was obtained from https://gadm.org/download_country_v2.html.

expected values (i.e. $f_{a,l}$). Moreover, the allocation algorithm ensures that individual middlemen deliver chickens to a desired number of LBMs, as specified by some statistical distribution. The agreement between empirical and simulated frequencies of unique LBMs visited daily is shown in Fig 2E. At the market level, wholesaling activities and vendor movements between LBMs further contribute to poultry mixing. Once a chicken enters an LBM, it may be sold multiple times to secondary vendors before reaching end-point consumers [18, 43]. In order to better capture the inner organization of LBMs, the model structures vendors in tiers according to their position along transaction chains (Fig 2F). Finally, we show the realised distribution of poultry marketing times alongside another estimate obtained using a different approach [18] (Fig 2G). Further statistics about individual actors and poultry transactions can be found in S1, S2 and S3 Figs.

Selected aspects of generated PDNs can be easily manipulated within our framework, allowing flexibility in exploring PDN configurations. In Fig 3, for example, we examine different distributions of LBMs serviced ($Pr(k_m)$) by individual middlemen on a daily basis (Fig 3A). As we increase the number of LBMs serviced per middleman, $\langle k_m \rangle$ on average, middlemen trade with more vendors (Fig 3B); consequently, individual transactions involve fewer birds since the total cargo is the same (Fig 3C). Fig 3B also suggests that the small discrepancy observed in Fig 3A at larger $\langle k_m \rangle$ is due to the limited amount of vendors (inset).

We also present the impact of vendors' trading practices on poultry marketing time. In particular, we alter the probability $p_{empty}$ that a vendor sells its entire cargo in a single day, the fraction $\rho_{unsold}$ of unsold birds in presence of some surplus (occurring with probability $1 - p_{empty}$). In addition, we consider high and low tendency to prioritise selling older (i.e. previously unsold) chickens over newly purchased ones. Varying parameters $\rho_{unsold}$ and $p_{empty}$ affects the average marketing time (Fig 4A), as well as the proportion of chickens being offered for sale on multiple days (Fig 4B). Full distributions of marketing times can be found in S4 Fig. Prioritizing the sale of older chickens had a negligible effect on these statistics. Indeed, prioritizing older chickens is compensated by a delay in selling newly purchased chickens (Fig 4C).

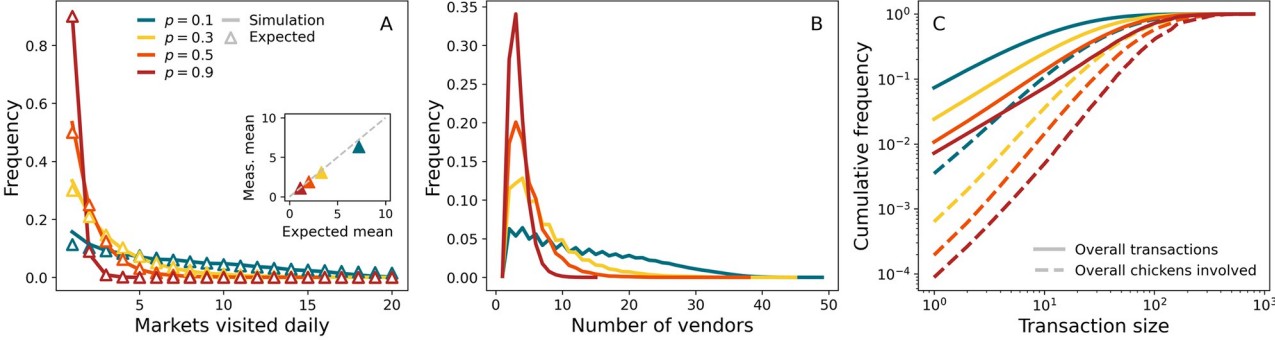

**Fig 3. Markets serviced daily.** (A) Empirical (scatters) and simulated (lines) distributions of markets serviced daily. Empirical distributions are of the form $Pr(k_m = n) \propto (1 - p_{k_m})^{n-1} p_{k_m}$ where $n = 1, \ldots, 20$. The inset compares empirical and simulated average numbers of markets serviced. (B) Distribution of vendors a single middleman trades daily with. (C) Cumulative distribution of sizes of transactions involving middlemen and vendors (solid lines). Dashed lines represent cumulative proportion of chickens sold in transactions up to a given size. Results are averaged over 50 simulations from 10 different PDN realisations.

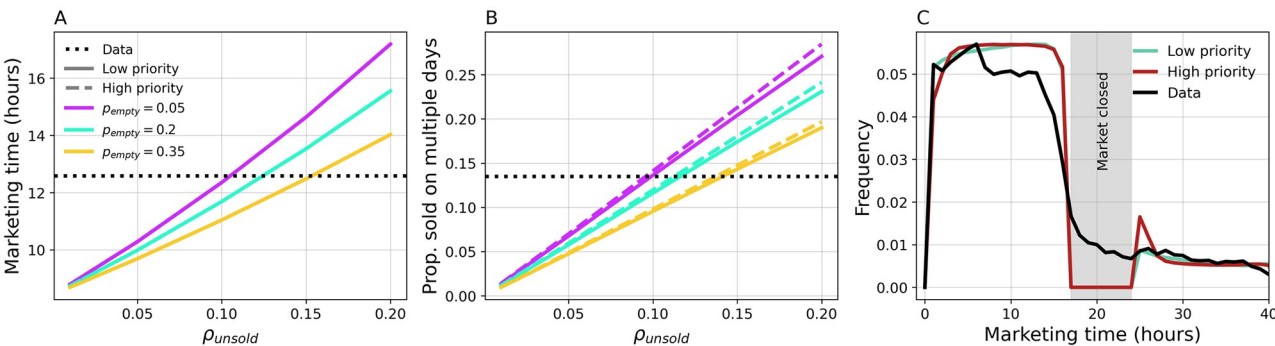

**Fig 4. Vendor trading practices.** (A) Average marketing time as a function of $\rho_{unsold}$ for different values of $p_{empty}$. Solid and dashed lines correspond respectively to low (10%) and high (90%) frequency of vendors prioritizing trading older chickens. (B) Proportion of marketed chickens offered for sale on multiple days. (C) Marketing time distributions for low and high frequency of vendors prioritizing older chickens. Here, $\rho_{unsold} = 0.1$ and $p_{empty} = 0.2$. Results are averaged over 50 simulations from 10 different PDN realisations.

As a final example, we examine the role of vendor movements between LBMs in promoting the mixing of chickens from different upazilas/sub-districts. Networks of LBMs defined by trader movements can vary considerably across poultry types, countries, and even cities within the same country [40]. In Chattogram, for example, vendors trading broiler chickens operate almost exclusively in a single market (Fig 5A). In Dhaka, however, this is not the case, resulting in frequent vendor movements that are articulated in a top-down structure where central and peripheral markets can be identified (Fig 5B). In fact, removing a single edge in the network shown in Fig 5B is sufficient to make it acyclic, suggesting a hierarchical organisation.

Within our framework, we encode inter-market mobility in a graph $G$, whose entries $G_{i,j}$ represent the probability that a vendor purchasing in market $i$ moves to market $j$ (or remains in $i$) to sell. As outlined above, vendors are further arranged in tiers, so that vendors in tier $L$ ($V_L$) can only buy poultry from wholesalers located in tier $L - 1$ or, in the case of $L = 0$ vendors ($V_0$) from middlemen trading in LBM $i$. For each vendor, purchase and sell locations remain fixed throughout a simulation.

To explore inter-market mobility, we use a generative network model to create mobility networks $G$ akin to that of Fig 5B. In practice, we generate directed acyclic graphs (DAGs) of varying density and amount of hierarchy (see Materials and methods section), according to parameters $\rho$ and $p_{random}$. $\rho$ represents the density of connections, while $p_{random}$ is the probability of an individual connection emanating from a source LBM that is selected randomly, rather than proportionally to their actual number of connections. To quantify the degree of hierarchy in a DAG $G$, we measure its global reaching centrality (GRC, see Fig 5C and 5E) and tree depth (TD, see Fig 5D and 5F). GRC measures how well every node can reach other nodes in the network with respect to the most influential node; it takes value 1 in the case of a star graph and approaches 0 when all nodes are equally influential (no hierarchy). In contrast, TD represents the longest directed path in $G$. Hierarchical DAGs, e.g. stars, tend to be more compact and hence shallower than random structures. Setting $p_{random} = 1$ yields DAGs with little hierarchy, as edges are allocated randomly. In contrast, $p_{random} \rightarrow 0$ introduces additional structure. Fig 5C and 5D show GRC and TD, respectively, for Dhaka's network and for DAGs generated with $p_{random} = 1$ (red) and $p_{random} = 0.1$ (cyan) while keeping the density of edges constant. Clearly, Dhaka's network is significantly more hierarchical and compact than random DAGs; in contrast, DAGs generated with $p_{random} = 0.1$ provide a much closer fit in terms of both GRC and TD.

We also summarize framework output by quantifying poultry mixing across 20 LBMs for different combinations of $\rho$ and $p_{random}$ (GRC and TD are shown in Fig 5E and 5F,

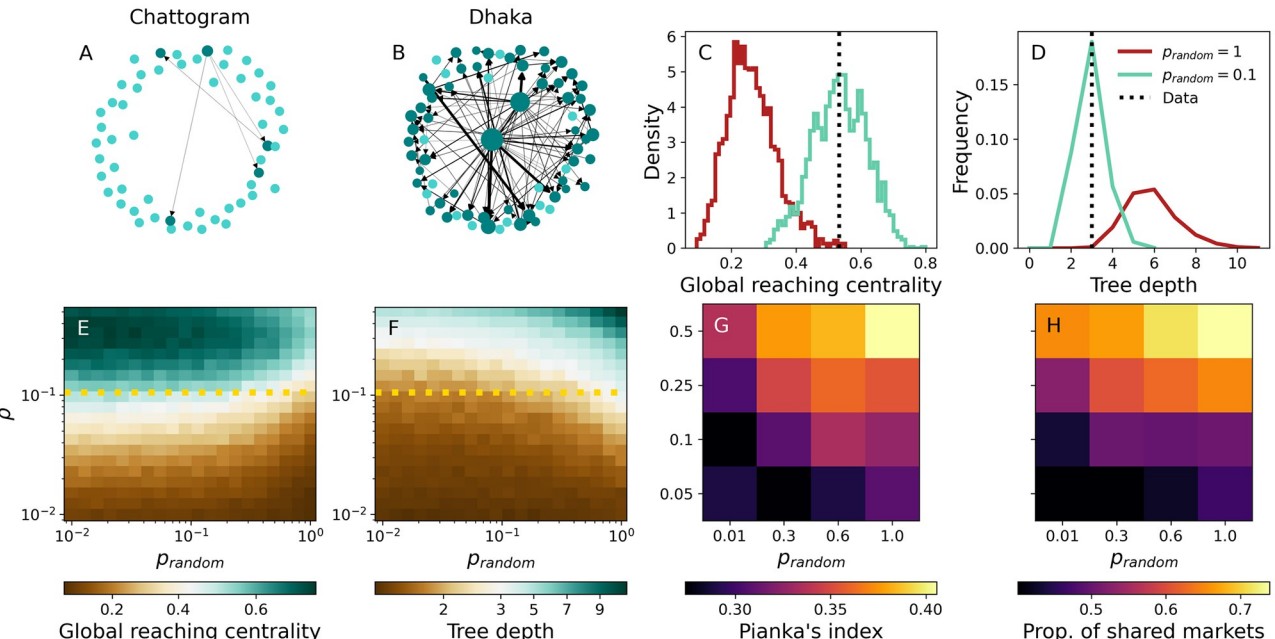

**Fig 5. LBM networks and poultry mixing.** (A,B) Broiler LBM networks for Chattogram and Dhaka, respectively. An arrow pointing from market $l$ to $l'$ indicates at least one movement in that direction, while arrow thickness is proportional to the number of vendors moving on that edge. Node size is proportional to the outgoing weight, i.e. the total number of vendors leaving it. Isolated and connected nodes are shown in cyan and teal, respectively. (C,D) GRC and TD, respectively, for Dhaka's network (line) and ensembles of 2000 synthetic LBM networks with the same density as Dhaka's network and $p_{random} = 1$ (red) and $p_{random} = 0.1$ (cyan). (E,F) Average GRC and TD, respectively, across 100 networks with 20 nodes and as a function of $\rho$ and $p_{random}$. The dotted line denotes Dhaka's density. (G,H) Pianka's index of overlap and proportion of markets where it is possible to find chickens from different upazilas/sub-districts, respectively, as a function of network parameters. Performing the same measurement before any vendor movement occurs, yields an overlap (Pianka's) of 0.261, and 25,7% shared markets, on average. This represents the baseline overlap due to middlemen sourcing chickens from farms and selling them to vendors. Results are averaged over 50 simulations from 10 different PDN realisations. All other PDN parameters are set to default values.

respectively). Here mixing refers to the extent to which chickens from distinct regions are brought together within LBMs. Upstream distribution, managed by middleman, and vendor movements between LBMs are the factors driving chicken mixing within this model. To quantify the amount of mixing, we record the geographic origins of chickens offered for sale in each LBM and use Pianka's index [44] to make pairwise comparisons of poultry populations marketed in distinct LBMs. Mean Pianka's index values are shown in Fig 5G as a function of parameters $\rho$ and $p_{random}$. Values close to 0 imply low overlap, while a value of 1 corresponds to identical distributions of geographic sources of poultry. Fig 5H shows another, complementary quantification of poultry mixing in terms of the mean number of LBMs where it is possible to find chickens from two randomly chosen upazilas/sub-districts. In general, we find that chicken mixing increases with network density, while hierarchy has the opposite effect: random vendor movements are more effective at mixing chickens within this simplified network model. It should be noted that high levels of mixing can be observed even in the absence of vendor movements due to upstream distribution (overlap between the catchment areas of LBMs, of which middlemen are responsible; see caption of Fig 5 for further details).

## Epidemic dynamics

In this section, we illustrate how our framework can be used to simulate and characterise pathogen transmission across PDNs. We first consider a single, AIV-like pathogen whose

dynamics is described by a Susceptible-Exposed-Infectious-Recovered (SEIR) model, as depicted in Fig 1B: upon infection, susceptible (S) chickens enter an intermediate exposed stage (E) and become infectious (I) after a short latent period $T_E$ = 6 hours. Infectious chickens recover (R) after an infectious period $T_I$ = 48 hours and become immune to further infection. Importantly, we assume that infected chickens do not die due to the disease and that sick chickens are not removed from LBM stalls. This assumption is broadly compatible with the epidemiology of low-pathogenic AIV strains such as H9N2 AIV, which are highly prevalent in Bangladeshi LBMs and only cause mild to sub-clinical symptoms in chickens [25–27]. We assume that the pathogen (repeatedly) emerges at rate $\alpha$ in farms due to external factors (e.g. contacts with wild birds) and spreads through the PDN through a combination of poultry movements, within- and inter-farm transmission (see Materials and methods section).

Model output comes at different levels of aggregation. Fig 6A shows for example daily incidence within LBMs during the first stages of an outbreak. At the most granular level, individual transmission events and their metadata can be tracked as well. Using this information, we can reconstruct transmission chains originating from individual introduction events and characterise their spatio-temporal evolution (Fig 6B). Fig 6C further characterises farm outbreaks by summarising attack rates by production cycle.

In Fig 6D–6F, we investigate the role of spatial transmission in an endemic context. We do so in a scenario where most transmission events occur within farms (Fig 6D), while viral amplification in LBMs is limited (Fig 6E). Here, spatial transmission is a crucial factor in determining global levels of infection. In the model, the intensity of inter-farm transmission is proportional to a spatial kernel $K(d)$ decaying with spatial distance $d$ and such that $K(0) = \beta_{FF}$. The latter parameter represents the maximum extent of inter-farm transmission at $d = 0$ (see also S1 Text and Table F therein). Increasing $\beta_{FF}$ facilitates spatial invasions, thus leading to more outbreaks in farms and more infections (Fig 6D). This results in an increasing number of infected chickens pouring into LBMs from farms (Fig 6F), explaining also the increase in within-market prevalence observed in Fig 6E.

Another important epidemiological question is whether AIV is transmitted and maintained in LBMs despite short marketing times. We address this question by considering an alternative endemic scenario where transmission is contributed mostly by LBMs (Fig 6G). We find that a major limiting factor to viral amplification in LBMs is represented by the latent period $T_E$ (Fig 6H): delaying the onset of infectiousness corresponds to a shorter window of opportunity for transmission under short marketing times. In order to further demonstrate this point, we quantify the persistence of transmission chains within LBMs (Fig 6I). As $T_E$ increases, opportunities for transmission are diminished and chains of infection stutter, leading to reduced persistence. In this case, the presence of AIV in LBMs can only be maintained through repeated introductions of infected poultry.

## Simulating multi-strain pathogens

Genomic surveillance in LBMs routinely identifies AIV lineages with distinct genetic signatures [45]. In some instances, the presence of multiple AIV subtypes, including the highly pathogenic H5N1 AIV, is also reported. Understanding this diversity requires, however, accounting for multiple, potentially interacting strains/pathogens that co-circulate in the same PDN. In this section, we use our framework to perform multi-strain simulations in a variety of PDN structures.

We illustrate this in Fig 7, which shows SEIR simulations with 50 co-circulating strains. For simplicity, we assume that these share the same epidemiological parameters, namely $T_E$, $T_I$ and $\beta$, and generate partial cross-immunity after a single infection.

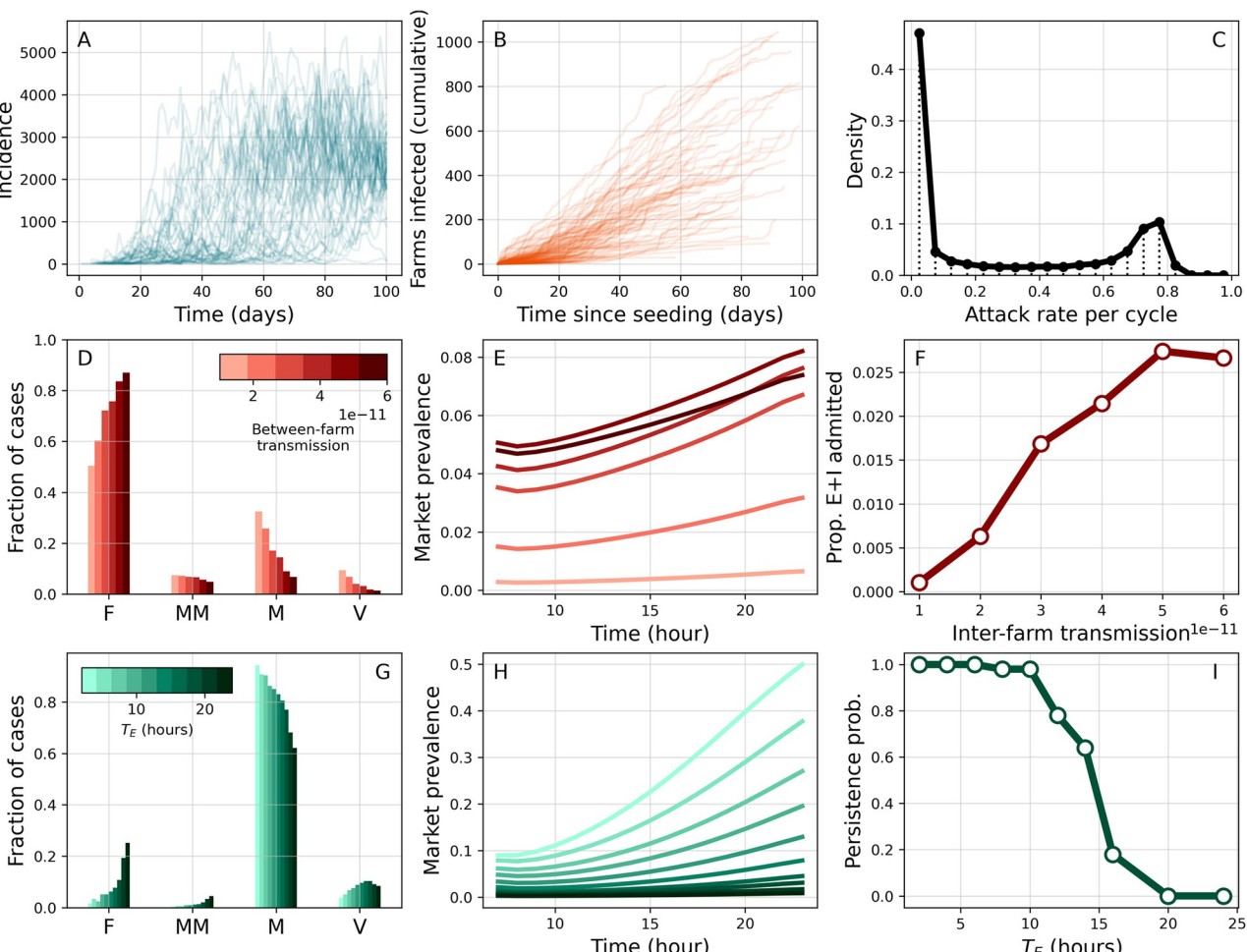

**Fig 6. Epidemic dynamics.** (A) Daily incidence in LBMs in multiple simulations. (B) Cumulative number of new farms infected over time from multiple clusters. Each cluster is initiated by a different infectious seed. (C) Distribution of attack rates for individual production cycles, conditional on at least one infection. (D-F) High farm transmission scenario ($w_F = 0.2$, $w_M = 0.7$). Colour scale corresponds to varying levels of inter-farm transmission $\beta_{FF}$. (D) Proportion of incident cases in different setting types (F: farms, MM: middlemen, M: markets, V: vendors). (E) Average hourly prevalence in LBMs at stationariety. (F) Proportion of latent and infectious chickens entering markets daily as a function of $\beta_{FF}$. (G-I) High LBM transmission scenario ($w_F = 0.1$, $w_M = 2.4$). Colour scale corresponds to varying latent period $T_E$. (G,H) mirror (D,E). (I) Persistence is measured as the proportion of simulations where at least one transmission chain persisting in markets and vendors for longer than 50 days was observed. Results are qualitatively the same under different criteria about the duration of transmission chains (S5 Fig). Other parameters are set to default values. Results are based on 50 simulations from 10 different synthetic PDNs.

Our aim is to measure the extent to which PDNs mix viral lineages from distinct geographical regions. To this end, we modify the external seeding protocol so that strain $s_i$, $i = 1, \ldots, 50$ can emerge only from upazila $i$. First, we investigate the role of viral amplification during the transport segment, which is operated by middlemen (Fig 7A–7C). To better disentangle the role of these actors, we consider low within-farm transmission and prevent inter-farm transmission fully by setting $\beta_{FF} = 0$. Consequently, viral mixing can not occur until chickens from different upazilas/sub-districts are collected by a middleman. As shown in Fig 7A, increasing transmission during transport by varying $w_{MM}$ leads to more infected chickens being introduced in LBMs, i.e. it results in viral amplification. Note, however, that below a certain value of $w_{MM}$, middlemen may introduce fewer infections in LBMs than those they picked up at farms. Increasing $w_{MM}$ has a modest positive effect on the average number of strains

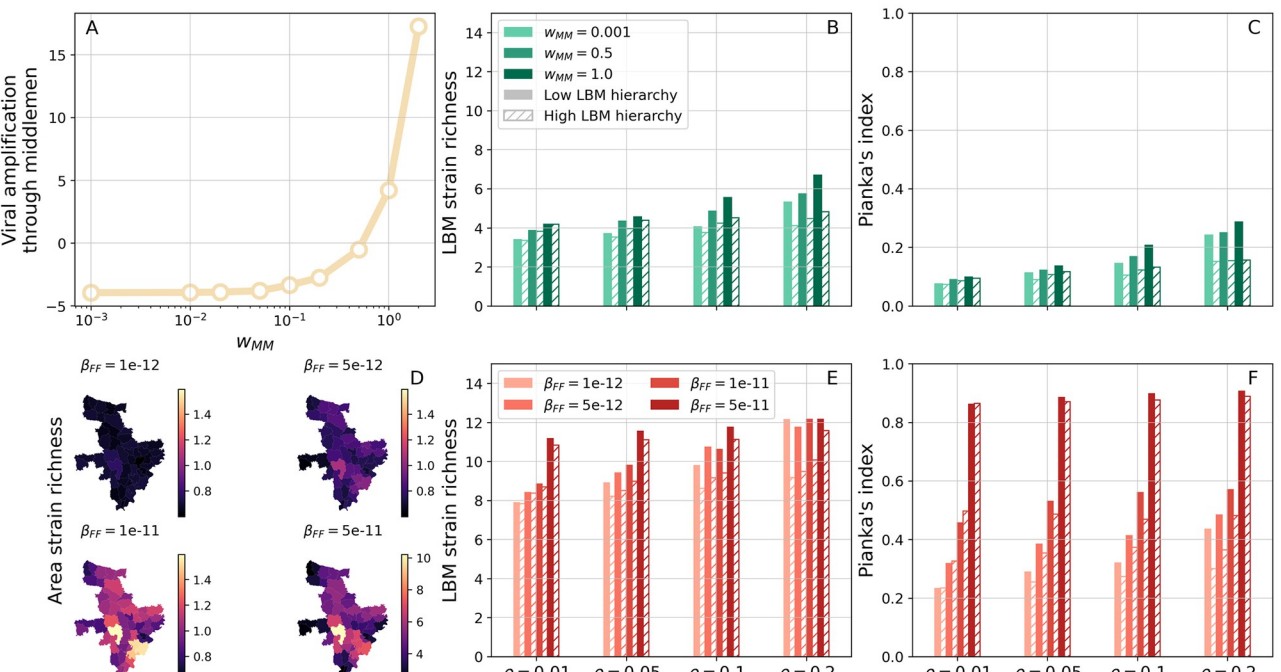

**Fig 7. Multi-strain dynamics and viral mixing in LBMs.** (A-C) Simulations with no inter-farm transmission ($\beta_{FF} = 0$). (A) Viral amplification happening through transportation from farm to LBM gates as a function of middlemen-specific transmission weight $w_{MM}$. This is quantified through the difference between total numbers of exposed and infected chickens sold to vendors and purchased daily by middlemen. (B) Average strain richness (i.e. number of strains) in single LBMs as a function of density $\rho$ of vendor movements (on the x-axis), $w_{MM}$ (from light to dark). Solid and striped bars correspond to low and high hierarchy in vendor movements, respectively. (C) Average Pianka's index of overlap between pairs of LBMs in terms of their catchment areas. (D-F) Simulations with inter-farm transmission. (D) Average richness per upazila for increasing $\beta_{FF}$. Note that the bottom-right map uses a different colour scale. (E,F) Same as (B,C) but for varying $\beta_{FF}$ and with $w_{MM} = 0.001$. We set $w_F = 0.1$ in (A-C) and $w_F = 0.2$ in (D-F), while $w_M = 2.4$ and $w_V = 1$ in all panels. Cross-immunity reduces susceptibility to secondary infections to $\sigma = 0.3$. Results are averaged over 50 simulations from 10 different synthetic PDNs. In each simulation, statistics are collected for 100 days after an initial transient of 2000 days. The base layer of the maps was obtained from https://gadm.org/download_country_v2.html.

circulating (i.e. strain richness) in individual LBMs, and on the overlap between LBMs in terms of circulating strains (light to dark bars in Fig 7B and 7C, respectively). We also explored, for fixed $w_{MM}$, the role of inter-market mobility on these metrics. In this context, the density $\rho$ of vendor movements had a positive effect on both strain richness and overlap between LBMs as it is promoting the dissemination of multiple strains across LBMs. In contrast, a more hierarchical layout of vendor movements decreased strain richness and overlap across LBMs. This is consistent with findings from Fig 5, which are suggestive of random inter-market connections promoting chicken's access to multiple LBMs. These results highlight the importance of accounting for inter-market movements to analyse the mixing of co-circulating AIVs.

Finally, we consider a further scenario in which transmission within and between farms plays a central role in shaping epidemic dynamics, while transmission occurring during transport is assumed to be negligible ($w_{MM} = 0.001$). We find that increasing inter-farm transmission $\beta_{FF}$ leads to a wider spatial dissemination of strains even outside their upazila of origin (Fig 7D). Consequently, a more diverse set of strains is supplied to LBMs, as evidenced by the number of strains observed at these locations (Fig 7E). Also, because larger values of $\beta_{FF}$ promote strain dispersal across the entire area, LBMs are now more similar to each other in terms of their strain populations (Fig 7F). It should be noted, however, that increased within-farm

transmission is responsible, at least in part, for the larger strain numbers and overlap between LBMs observed in Fig 7E and 7F with respect to panels B,C. Finally, we note that the effects of density and hierarchy of vendor movements on ecological metrics are analogous to those observed in the previous scenario. These results are robust to increasing cross-immunity between strains (S6 Fig).

## Discussion

In this paper we have introduced EPINEST, an agent-based model to simulate the transmission of generic health hazards in the context of realistic poultry or livestock movements within a defined PDN. To the best of our knowledge, this work represents the first attempt to account for the structural complexities of poultry PDNs in the context of epidemic transmission modelling. Our model allows to generate synthetic PDNs consisting of key actors and settings involved in poultry production and distribution, namely farms, middlemen, LBMs and market vendors. Using Bangladesh as a case study, we illustrated the ability of our framework to reproduce empirical features of a broiler PDN. We used extensive data from field surveys to inform most aspects of the model, including farming and trading practices of key actors [18, 27, 46]. At the same time, our model offers the possibility to easily manipulate most properties of the network, allowing exploration of alternative PDN configurations. Importantly, we emphasize that our model may be applied to other contexts, e.g. different poultry types and countries for which sufficient data is available.

One of the main purposes of EPINEST was to assess the impact of PDN structure and stakeholders' trading practices on pathogen transmission. For this reason, we prioritised including PDN components with the highest relevance to transmission dynamics. These include, for example, the time spent by chickens at different locations. All-in/all-out production, which is commonly implemented in commercial broiler farms, results in relatively homogeneous rearing times across farmed chickens, although these vary considerably among different chicken types. In contrast, LBMs are characterised by a much faster turnover, with most chickens being sold within a few hours and unsold chickens remaining for up to a few days. Longer marketing times are a well-established risk factor for AIV infection in LBMs, and have been linked to AIV persistence in these settings [18, 47]. To account for heterogeneity in marketing times, we explicitly account for a fraction of chickens being offered for sale on consecutive days.

Further basic ingredients of the model are the spatial distribution of poultry farms and their sizes. Both elements are highly relevant to disease transmission. Heterogeneities in farm locations can affect systemic vulnerability to epidemics and pathogen dispersal patterns [48–50], while higher livestock densities are associated with increased intra-farm transmission and may favor the emergence of virulent pathogens [9, 51]. In the absence of accurate data about farm locations, we generated random farm distributions complying with reported volumes of poultry production at the upazila level, and used field surveys to assign farm sizes [18, 27]. Nonetheless, we stress that our model can accommodate any distribution of farms. These may represent not only higher-resolution data, but also outcomes from more accurate generative models [52–54].

Our model also allows to control the degree of mixing of chickens along distribution and trading channels. The ability of PDNs to mix large numbers of chickens, particularly within LBMs, is well-established. The inter-mingling of different types of birds from potentially distant locations is concerning when associated to co-circulation of genetically distinct viruses. A recent phylodynamics study found substantial genetic structuring of H9N2 AIV by city in Bangladesh [55], compatibly with low overlap between the corresponding supplying production

areas [18]. In contrast, viral lineages appeared to be highly mixed across LBMs within the same city, possibly indicating frequent connections between these markets. Live poultry trade has also been shown to be an important driver of regional AIV dissemination in China [56]. Here, poultry mixing within and between LBMs is dictated by two factors: first, upstream distribution via middlemen connects LBMs with farmed populations from a wide geographic area. Within our framework, geographic fluxes between regions (upazilas/sub-districts) and LBMs are expressed as a matrix that can be informed using field surveys or traceability systems. Second, wholesaling activities and vendor movements further contribute to stirring marketed poultry across LBMs. In this manuscript, we used a generative model to sample inter-market mobility networks, and quantified their impact on poultry mixing. We emphasize that more complex mobility patterns, informed either from data or through simulations, can be easily embedded within our framework.

A major feature of our model is that it allows simulating pathogen transmission while accounting for the complexity of poultry movements and PDN structures. Importantly, the epidemic layer is fully uncoupled from PDN generation. Thus, while current code supports simulations of SIR and SEIR dynamics only, implementing additional epidemic models is a relatively straightforward task. We illustrated how our ABM can be used to model AIV dynamics in both epidemic and endemic settings. In the former case, the ABM makes it possible to map the early dissemination of, e.g., an emerging AIV strain across farms and the rest of the PDN. The second scenario would be more suitable to describe endemic circulation of AIVs. In this context, relevant scientific questions that could be addressed using our framework include understanding how and where an endemic AIV is maintained and amplified along the PDN.

A novel aspect of our ABM is that it enables simulations of multiple co-circulating pathogens/strains and their interactions. This paves the way for a number of eco-epidemiological applications. As an example, we assessed the potential of PDNs to mix viral lineages originating from distinct geographical areas. Additional applications may consider the joint dynamics of endemic and emerging AIVs and simulate the early transmission dynamics of, say, highly-pathogenic H5N1 AIV against a background of (cross-)immunity generated by endemic circulation of H9N2 AIV [57].

As any modelling framework, there are limitations to our ABM. Despite our efforts to account for the structural complexity of PDNs, our focus on epidemiological investigations meant that several aspects of real PDNs could not be included in the model. For example, actors' behaviours are treated as fixed parameters external to, rather than emerging from, the dynamic system being modelled. In reality, the decisions made by individual traders to sell or purchase birds is influenced by social, economic and epidemiological factors. These may include uncertainty about market conditions and fear spurred by disease outbreaks [58, 59]. In addition, unequal power dynamics often constrains trading ties [40, 60]. In this context, we plan on expanding our ABM's capabilities to include simple reactive behaviours, e.g. farmers selling chickens pre-emptively following a surge in bird mortality [59]. An important assumption in EPINEST is that chickens mix homogeneously within a given setting. This simplification may be less accurate for larger farms that store multiple flocks in separate sheds, or for large villages where multiple backyard poultry flocks are raised. Heterogeneity in local transmission could be partly recovered by 'splitting' large farms in smaller, neighbouring units and by exploiting inter-farm transmission (as if different units were distinct farms). In its current implementation, EPINEST assumes pathogens to spread directly from infected to susceptible chickens. However, environmental contamination, caused by infected chickens, can also affect transmission and contribute to pathogen dispersal within and between sites, especially from LBMs back to farms via contaminated mobile traders. Future versions of this software may

incorporate these additional routes of transmission. Further extensions to the model could include mixing of different poultry species, different farming systems and trading practices, such as second-line middlemen purchasing chickens from other traders, and different biosecurity measures implemented at different LBMs to limit pathogen spread. Finally, although we wrote our model in C++ to improve simulation speed (An assessment of simulation times is shown in S7 Fig), computational constraints make it difficult to scale up simulations to more than a few millions of farmed chickens. This is a common challenge in agent-based models, where the increased amount of detail is traded off by computational costs.

This ABM framework would allow users to conduct a more comprehensive assessment of intervention effectiveness, thereby yielding more meaningful recommendations. For interventions classically implemented in other disease transmission models such as culling and vaccination, for example, the ABM enables the evaluation of their impact across the entire PDN. Moreover, the model facilitates the exploration of interventions that seek to modify the PDN configuration—such as rewiring the network of mobile traders and/or vendors—an aspect that has been overlooked in the modelling literature despite its relevance for mitigating disease risks.

In conclusion, we implemented a novel agent-based model to jointly simulate realistic poultry movements and epidemic trajectories. Realised structures encompass a wide-range of PDN configurations as encountered in many countries in South and Southeast Asia, and potentially even other livestock production systems with similar structure to the one discussed here. Compared to existing ABMs devoted to veterinary epidemiology applications [61–64], ours offers the ability to run both single- and multi-strain simulations. In addition, the simulator can be programmed to yield a wide range of outputs, including individual transactions and chains of infections, hence providing a full characterisation of the underlying system. This model is a unique tool in the One Health context as it allows investigation of a range of epidemiological scenarios and helps us to understand better the role of different structural aspects on disease transmission. Immediate applications of this model will allow exploration of the transmission and amplification of AIVs and anti-microbial resistance genes within poultry PDNs.

## Materials and methods

### Generating synthetic PDNs

In general, a PDN denotes the ensemble of actors that are involved in the production and/or distribution of a product such as poultry and their interactions. At any point in time, a chicken is physically located within one and only one setting, such as a farm, a middleman's truck, an LBM or a vendor-owned shed during the night.

Our generative algorithm instantiates a population of actors based on external specifications. First, a spatial distribution of farms must be provided alongside the corresponding geographic setup. The latter consists of a partitioning of the study area into a set of non-overlapping regions. In this study, we take upazilas/sub-districts as regional units. Second, the user specifies a number of LBMs and their catchment areas. In practice, this is achieved by specifying a matrix $f_{a,l}$ representing the relative fluxes of chickens reaching market $l = 1, \ldots, N_M$ from area $a = 1, \ldots, N_A$. A full description of LBMs requires a set of weights $G_{l,l'}$ encoding the probability that a vendor purchases chickens in LBM $l$ and trades in LBM $l'$ (with possibly $l = l'$). Finally, a number of parameters influencing farming, distribution and trading practices should be specified as well (these are described in S1 Text). With these details, the algorithm computes the expected poultry fluxes between farms and LBMs and allocates enough vendors and middlemen to satisfy such demand. At the LBM stage, vendors are allocated in a tier-wise fashion depending on the volume of chickens supplied by middlemen, inter-market

movements, and wholesaling practices. Eventually, it is possible to generate more middlemen and vendors than strictly required based on heuristic calculations by inflating the expected supply of chickens handled by middlemen and vendors through multiplicative factors $\epsilon_{MM}$ and $\epsilon_V$.

## Modelling inter-market movements

As detailed in S1 Text, a vendor purchasing chickens in LBM $i$ is assigned to trade in LBM $j$ with probability $G_{i,j}$.

We sample the weights $G_{i,j}$ from a generative network model defined by a growth mechanism: we add LBMs $j = 1, \ldots, N_M$ one at a time and establish links $i \rightarrow j$, $i < j$ as follows: first, we draw the number of incoming edges (in-degree) $z_j \sim Binomial(\rho, j - 1)$. Second, we sample $z_j$ LBMs (with $i < j$) without replacement either at random with probability $p_{random}$, or proportionally to their current out-degree. As nodes acquire more connections, they increase their ability to attract further links whenever $p_{random} < 1$. $p_{random} = 1$ completely suppresses the advantage of nodes with larger out-degrees as it assigns edges completely at random. This process yields a network topology characterised by a binary matrix $\mathcal{I}_{i \rightarrow j}$ that denotes existing, oriented connections from node $i$ to $j$ ($\mathcal{I}_{i \rightarrow j} = 1$). Within the fully-grown network, the out-degree of node $i$ is calculated as $k_i = \sum_j \mathcal{I}_{i \rightarrow j}$. Weights $G_{i,j}$ are then calculated as:

$$G_{i,j} = \begin{cases} \delta_{i,j}, & \text{if } k_i = 0. \\ G_{self}\delta_{i,j} + (1 - G_{self})\mathcal{I}_{i \rightarrow j}/k_i, & \text{otherwise}, \end{cases} \quad (1)$$

where $\delta_{i,j}$ is Kronecker delta, which is 1 if $i = j$ and is 0 otherwise, and $G_{self}$ is the probability of a vendor operating in a single market $i$, conditional on the out-degree $k_i$ being positive. Here we set $G_{self} = 0.8$.

## Global reaching centrality

GRC is defined based on the notion of local reaching centrality $C_R(i)$, which quantifies the proportion of nodes reachable from node $i$ through directed edges. Based on this definition, we calculate GRC by subtracting $C_R(i)$ from the maximum observed value $C_R^{max} = \max_i C_R(i)$, and averaging over all nodes:

$$GRC = \frac{\sum_i^N C_R^{max} - C_R(i)}{N - 1} \; . \quad (2)$$

## PDN dynamics

Within simulations, actors follow a daily routine. Let $t = 0, \ldots, 23$ indicate the time of the day (each time step is 1 hour long). Unless otherwise stated, default parameters indicated in Tables A-E in S1 Text are considered. LBMs open between $T_{open}$ and $T_{close}$; at $t = T_{open}$, vendors move to LBMs, followed by middlemen. Middlemen then proceed to sell their cargo to frontline vendors, i.e. those in the first LBM tier ($L = 0$). In the next time step ($t = T_{open} + 1$), some of these move to another LBM and trade with second-tier vendors, who in turn sell chickens to vendors in the tier after that, repeating the process until the last tier is reached. Vendor movements and wholesaling are therefore resolved sequentially, in a tier-wise fashion, at time $t = T_{open} + 1$. In contrast, retailing activities roll out between $T_{open} + 1$ and $T_{close}$. At $T_{close}$, both wholesalers and retailers leave LBMs alongside any unsold chickens. Overnight, these chickens are stored in

some other place, e.g. in a shed. Importantly, all chickens from the same vendor are stored in the same place.

At some time $t = T_{farm}$ we update farms: empty farms may recruit a new batch of chickens, while active farms may offer birds for sale depending on batch age. After that, always at $t = T_{farm}$, middlemen are updated: first, they decide whether to cover a different set of upazilas/sub-districts. Then, they contact farms within covered upazilas/sub-districts in order to purchase chickens. At this stage, middlemen only determine how many chickens to collect from each farm; the collection may happen anytime between $T_{farm}$ and $T_{open}$ on the following day.

### Epidemic dynamics

In this work, we consider a general transmission model involving a generic number of strains. Each strain, indexed by $s$, spreads according to SEIR dynamics. Infected chickens become infectious only after a random latent period $\hat{\tau}_E$, sampled from a distribution $P(\hat{\tau}_E)$ with mean $T_E$. Analogously, infectious chickens recover after a random time $\hat{\tau}_I$, sampled from a distribution $P(\hat{\tau}_I)$ with mean $T_I$. All epidemiological parameters are listed in Table F in S1 Text.

Here, transmission is assumed to occur through infectious contacts among chickens from the same setting. Other transmission mechanisms, including external introductions and inter-farm transmission, are described in S1 Text. During a time step, an infectious chicken $i$ contacts a single chicken $j$, chosen at random within the same setting, and transmits strain $s$ with probability:

$$p_{infect} = 1 - exp(-\beta(s, i) \cdot S(s, j) \cdot w_X) , \qquad (3)$$

where $\beta(s, i)$ denotes the transmissibility of strain $s$ and $S(x, j)$ is the susceptibility of chicken $j$ to $s$. The factor $w_X$ is a multiplier that depends only on the underlying setting type (F,MM,M,V).

In general, the transmission rate $\beta(s, i)$ and susceptibility $S(s, j)$ may depend on the immune state of infector and infectee, respectively. Importantly, different functional forms of $\beta(s, i)$ and $S(s, j)$ embody different assumptions about immune cross-reactions induced by previous exposure to other pathogens/strains. In this work we consider uniform transmission $\beta(s, i) = \beta$, irrespective of immune state, and susceptibility $S(s, j) = 0, 1, \sigma$ depending on whether $j$ has already been infected with $s$, is fully naive or was infected with some other strain $s' \neq s$, respectively. The parameter $\sigma \in [0, 1]$ represents reduced susceptibility due to cross-immunity, and interpolates between sterilising cross-immunity ($\sigma = 0$) and no cross-immunity ($\sigma = 1$).

### Supporting information

**S1 Text. Supplementary methods.** File containing further details about data analysis as well as model simulation and initialisation.
(PDF)

**S1 Fig. Additional farm statistics from simulations.** (A) Distribution of numbers of production cycles completed per year. The simulated distribution (red) appears narrower compared to empirical data (black) [27]. However, it should be added that several interviewed farmers raised multiple batches simultaneously, and those that declared raising a single batch during the interview may well have being managing 2 or more simultaneously during the previous year. (B) Cumulative distribution of sizes of transactions involving farms and middlemen (solid line). The dotted line represents the cumulative proportion of chickens sold in transactions up to a given size. The corresponding distributions, denoted with $p_s$ and $p'_s$ respectively, are related since $p'_s = s \cdot p_s / \sum_s s \cdot p_s$. In other words, $p'_s$ is the size-biased version of $p_s$. (C) Proportion of chickens remaining unsold after a given time since being offered for sale for the first

time by a farmer. Note that it is highly unlikely for a chicken to remain unsold for more than 5 days. Results are obtained from a single simulation with default settings as in Fig 2 in the main manuscript.
(PNG)

**S2 Fig. Additional middlemen statistics from simulations.** (A) Proportion of upazilas visited daily by one middleman during a single simulation. Note that a middleman may visit up to 4 upazilas per day, but visiting one or two is usually sufficient to complete a cargo. (B) Distributions of daily numbers of farms visited by one middlemen (cyan) and middlemen visiting one farm (red). (C) Distribution of numbers of vendors trading daily with a middleman. (D) Cumulative distribution of sizes of transactions involving middlemen and vendors (solid line). The dotted line represents the cumulative proportion of chickens sold in transactions up to a given size. Note that these transactions are typically smaller than those between farms and middlemen (S1 Fig) since vendors deal with smaller amounts of chickens than other PDN actors. Results are obtained from a single simulation with default settings as in Fig 2 in the main manuscript.
(PNG)

**S3 Fig. Additional vendor statistics from simulations.** (A) Distribution of numbers of wholesalers supplying a retailer (yellow), another wholesaler (blue) or any vendor (red) on a daily basis. (B) Distribution of numbers of retailers (yellow), wholesalers (blue) or vendors (red), regardless of type, purchasing from a single wholesaler on a daily basis. Note that (A) excludes vendors buying chickens from middlemen, i.e. vendors operating in the first LBM tier. (C) Distributions of daily amounts of chickens bought from retailers (yellow) and wholesalers (blue) in simulations (lines) and data (markers) [18]. (D) Cumulative distribution of sizes of transactions involving middlemen and vendors (solid line). The dotted line represents the cumulative proportion of chickens sold in transactions up to a given size. Results are obtained from a single simulation with default settings as in Fig 2 in the main manuscript.
(PNG)

**S4 Fig. Distribution of marketing times.** Each panel shows distributions of marketing times for different average proportions of unsold chickens $\rho_{unsold}$ and for increasing probability $p_{empty}$ of a vendor selling all chickens in a single day (from left to right). The marketing time is defined as the time interval elapsed since a chicken enters any LBM for the first time and is sold to an end-point customer. Simulation settings are the same as in Fig 4 with only 10% of vendors prioritizing the sale of unsold chickens.
(PNG)

**S5 Fig. Sensitivity of persistence probability to duration of transmission chains.** Lines show how the probability of pathogen persistence varies with both $T_E$ and the minimum duration to determine whether a transmission chain is persistent or not. The estimation of the probability of persistence as well as simulation settings are the same as in Fig 6I.
(PNG)

**S6 Fig. Viral mixing under complete cross-immunity.** Results mirror panels B,C,E,F from Fig 7 in the main manuscript, under the assumption of complete cross-immunity ($\sigma = 0$). Increasing cross-immunity lowers strain richness in any setting as individual strains face increased competition. Nonetheless, increasing cross-immunity does not significantly affect overlap between LBMs.
(PNG)

**S7 Fig. Simulation time analysis.** (A) Mean simulation time (seconds) as a function of the number of farms and in absence of pathogen transmission. (B) Mean size of the chicken population as the number of farms increases. (C) Mean simulation time in presence of pathogen transmission for different combinations of within- and inter-farm transmission intensity (parameters $\beta_F = \beta \cdot w_F$ and $\beta_{FF}$, respectively). Here the number of farms is set to 1200. (D) Mean pathogen prevalence in farms (proportion of infected farmed chickens) as a function of the same parameters. Our findings indicate that simulation time increases linearly with the size of the poultry population and non-linearly with pathogen prevalence when parameters $\beta_F$ and $\beta_{FF}$ are varied. This happens because large values of prevalence can be achieved only if the pathogen is able to transmit sufficiently well both within and across poultry flocks. Results are averaged over 50 simulations from 5 different PDN realisations. In panels A,B, each simulation is run for 6 years. In panels C,D, each simulation is run for 8 years and the pathogen is introduced after 4 years. For simplicity, we consider transmission within settings other than farms to be negligible (we set $w_{MM} = w_M = w_V = 0.001$). Other PDN and epidemiological parameters are set to default values.
(PNG)

## Author Contributions

**Conceptualization:** Sunetra Gupta, Fiona Tomley, Guillaume Fournié.

**Formal analysis:** Francesco Pinotti.

**Funding acquisition:** Fiona Tomley, Guillaume Fournié.

**Investigation:** Suman Das Gupta, Joerg Henning, Tony Barnett, Dirk Pfeiffer, Md. Ahasanul Hoque, Guillaume Fournié.

**Methodology:** Francesco Pinotti.

**Project administration:** Sunetra Gupta, Fiona Tomley, Guillaume Fournié.

**Software:** Francesco Pinotti.

**Supervision:** José Lourenço, Sunetra Gupta, Guillaume Fournié.

**Visualization:** Francesco Pinotti.

**Writing – original draft:** Francesco Pinotti, José Lourenço, Sunetra Gupta, Suman Das Gupta, Joerg Henning, Damer Blake, Fiona Tomley, Tony Barnett, Dirk Pfeiffer, Md. Ahasanul Hoque, Guillaume Fournié.

**Writing – review & editing:** Francesco Pinotti, José Lourenço, Sunetra Gupta, Suman Das Gupta, Joerg Henning, Damer Blake, Fiona Tomley, Tony Barnett, Dirk Pfeiffer, Md. Ahasanul Hoque, Guillaume Fournié.

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
