## [Decision Letter · Decision Letter 0]

12 Dec 2023

Dear Dr Pinotti,

Thank you very much for submitting your manuscript "EPINEST, an agent-based model to simulate epidemic dynamics in large-scale poultry production and distribution networks" for consideration at PLOS Computational Biology. As with all papers reviewed by the journal, your manuscript was reviewed by members of the editorial board and by several independent reviewers. The reviewers appreciated the attention to an important topic. Based on the reviews, we are likely to accept this manuscript for publication, providing that you modify the manuscript according to the review recommendations.

Reviewers were impressed with the work's robustness and significance but suggest some further explanation and discussion for improvement. Please revise the manuscript according to these suggestions.

Sincerely,

Kim M. Pepin

Guest Editor

PLOS Computational Biology

Virginia Pitzer

Section Editor

PLOS Computational Biology

Reviewer's Responses to Questions

**Comments to the Authors:**

Reviewer #1: The authors present a very well written, well documented paper with an elaborate study of the effect of poultry farming and trading practices on pathogen spread, maintenance and evolution. The model is well explained in the main document and documented in detail in the supplementary information, so it is relatively easy for the reader to find model assumptions and parameter values. My comments are minor and given below.

General remarks:

• It would be good to mention in the abstract, introduction and also the title that this work concerns broilers, as the epidemic dynamics in layer hens would be very different.

• In supplementary material, line 464 it is mentioned that "We make the assumption that chickens within the same setting mix homogeneously at random". This is a valid choice for the modelling, but does not necessarily reflect reality, especially in very large farms. Can you spend some attention on this modelling choice in the discussion and explain in what ways the model could deviate from reality because of this choice.

• Chapter "Epidemic dynamics", line 197. At the LBMs, are infectious (I) chickens removed from the population, or are they allowed to recover (R)? I can't imagine that people want to buy sick, infected chickens.

Detailed remarks

• Figure 2, panel B.

○ The text mentions that black lines show expected values, while red lines are for measured values. The legend on the other hand assigns red to simulations and black to data. Which is correct?

○ Explain the abbreviation PMF in the figure legend. The legend should be readable on its own. Please check other figures too (include CMF).

○ How was the fit of the red lines assessed?

• Figure 7. " (D-G) Simulations with inter-farm..." should be " (D-F) Simulations with inter-farm...".

Reviewer #2: The article is well written and the work done is impressive . Just few very minor points that could improve the impact of the paper

The first time $\\beta_{FF} is introduced no definition is given. I would suggest to make a Table (maybe in the Supplementary Information) collecting all the parameters used , their definition and their estimates or sources

A second , maybe more important point, wold be to provide a table resuming the performances of the tool: comparing different scenarios, size of populations , network structures and sizes,and the time to run a set of simulations. This could be done as supplementary material

Third, I would suggest to change the color scale used in Figure 6 D_I,

Reviewer #3: 1. The article represents a significant model development effort and describes a novel ABM approach that enables capturing the complexities of disease transmission dynamics in poultry production and distribution networks. The modeling framework incorporates flexibility to evaluate specific PDN related questions informed by data as well as more open-ended questions and sensitivity analysis. In addition, the framework enables assessing the impact of network structure aspects (e.g., hierarchy in vendor transactions), mixing and pathogen amplification along the value chain on the disease transmission dynamics in live bird markets. The capability to model multiple co-circulating strains and strain diversity is a unique feature relative to existing models.

2. The current PDN modeling scenario does not consider the impact of environmental contamination in LBMs and potential farm exposures via vehicles or personnel returning from LBMS towards disease persistence and strain richness. I suggest discussing these as potential limitations as applicable.

3. Are there any recommendations for potential mitigations at various levels of the PDN to reduce AI persistence, prevalence and strain mixing at LBM based on the simulation results?

4. Line 10: typo such instead of “suck risk”

5. Line 262: LBMs. “In contrast, a larger degree of hierarchy in movements (striped bars) had the opposite effect, in agreement with findings from Fig. 5.” As the impact of hierarchy is a key insight, suggest elaborating this statement to improve clarity (i.e., impact of hierarchy on network reachability and overlap potentially contributing to reduced strain richness)

6. Line 418: Define δi,j used in equation 1 and how was it was estimated. Also, could you provide further explanation on how δi,j is positive when the outdegree ki is 0 and thus there are no connections between markets I and J?

7. Lines 414-418: Is the outdegree for each LBM informed by data or is it calculated and updated dynamically in the generative network model? Consider expanding this paragraph to explain the network growth mechanism in greater detail.

8. Probability that a vendor purchases in LBM l and sells in l’ is referred to as wl,l’ in the supplement (Table A2) and Gij in the main text (line 410). Are these two variables equivalent? If not, could you clarify the difference.

9. Line 22 in supplement: Why was a negative binomial distribution chosen for the replenishment time. Were any other statistical distributions explored?

10. Supplement Table A6: How was the value of βFF (5 * 10−11 days−1) chosen in the baseline scenario?

11. Supplement equation 12 and Table A6: Is the transmission kernel K(D) in the same form as in reference 6? If not provide details on how the parameters γK and Dk were obtained from the reference.

**Have the authors made all data and (if applicable) computational code underlying the findings in their manuscript fully available?**

Reviewer #1: Yes

Reviewer #2: Yes

Reviewer #3: Yes

PLOS authors have the option to publish the peer review history of their article (what does this mean?). If published, this will include your full peer review and any attached files.

Reviewer #1: No

Reviewer #2: No

Reviewer #3: No

Figure Files:

Data Requirements:

Reproducibility:

References:

---

## [Editor Report · Decision Letter 1]

6 Feb 2024

Dear Dr Pinotti,

We are pleased to inform you that your manuscript 'EPINEST, an agent-based model to simulate epidemic dynamics in large-scale poultry production and distribution networks' has been provisionally accepted for publication in PLOS Computational Biology.

Best regards,

Kim M. Pepin

Guest Editor

PLOS Computational Biology

Virginia Pitzer

Section Editor

PLOS Computational Biology

---

## [Editor Report · Acceptance letter]

16 Feb 2024

PCOMPBIOL-D-23-01184R1 

EPINEST, an agent-based model to simulate epidemic dynamics in large-scale poultry production and distribution networks

Dear Dr Pinotti,

I am pleased to inform you that your manuscript has been formally accepted for publication in PLOS Computational Biology. Your manuscript is now with our production department and you will be notified of the publication date in due course.

With kind regards,

Anita Estes
